# Learning to Plan for Language Modeling from Unlabeled Data

**Nathan Cornille, Marie-Francine Moens & Florian Mai**
Department of Computer Science
KU Leuven
Leuven, Belgium
{nathan.cornille,sien.moens,florian.mai}@kuleuven.be

## Abstract

By training to predict the next token in an unlabeled corpus, large language models learn to perform many tasks without any labeled data. However, their next-token-prediction objective arguably limits their performance in scenarios that require planning, such as writing a coherent article. In this paper, we train a module for planning the future writing process via a self-supervised learning objective. Given the textual context, this planning module learns to predict future abstract writing actions, which correspond to centroids in a clustered text embedding space. By conditioning on these actions, our model extends the successful language model formula to more abstract planning in an unsupervised way. Empirically, we demonstrate that our method improves language modeling performance in general, particularly with respect to the text structure. Because our framework uses a planner module that is unsupervised and external to the language model, new planner modules can be trained at large scale and easily be shared with the community.

## 1 Introduction

Despite their simple learning objective, language models (LMs) can solve a surprising range of challenging generation tasks such as summarization (Zhang et al., 2024b), style transfer (Reif et al., 2022), and machine translation (Zhu et al., 2023). This is widely attributed to the fact that the self-supervised next-token-prediction objective allows for training large models on unprecedented amounts of unlabeled data (Radford et al., 2019), unlocking new capabilities at sufficient scale (Wei et al., 2022a), e.g. in-context learning (Brown et al., 2020) and chain-of-thought reasoning (Wei et al., 2022b). The dual-process theory posits that human intelligence comprises intuitive (type 1) and deliberate (type 2) reasoning systems (Evans, 1984; Kahneman, 2011). Bengio et al. (2021) describe current self-supervised deep learning systems as most successful at system 1 tasks such as perception, reading, and talking. However, it is currently unclear how to transfer the success of self-supervision to system 2 tasks that require planning, such as writing a coherent article.

Most popular approaches for eliciting deliberate reasoning capabilities in LMs are based on either advanced prompting techniques such as chain-of-thought (Wei et al., 2022b), plan-and-solve (Wang et al., 2023a), or tree-of-thought (Long, 2023), or on finetuning on task-specific reasoning data (Havrilla et al., 2024). While these techniques already lead to substantial improvements, they ignore the key ingredient that made LMs so successful in the first place: the ability to obtain ever-improving performance through scalable self-supervised pretraining. Hence, it is widely recognized that combining planning with self-supervised foundation models is a promising research direction (Yang et al., 2023).

In the context of sequential decision making, planning is the process of determining a decision policy from a model of the environment (Sutton & Barto, 2018). In this paper, we propose a method for learning to plan future writing from unlabeled data. Concretely, as illustrated in Figure 1, we first transform unlabeled text into a sequence of abstract writing actions which we obtain from a clustered text embedding space. We then train an external planner network which predicts the next writing action from the current context. Finally,

we integrate the new learned planning capability into a regular LM by finetuning it on the self-supervised next-token-prediction task in a way that allows it to extract relevant information from predicted writing actions.[1]

In summary, we report the following findings:

1. Our model makes effective use of abstract writing actions predicted by the planner.
2. An external planner outperforms an LM-internal planner.
3. The LM improvements are specifically due to improvements in anticipating and matching the text structure due to planning.

## 2 Related Work

### 2.1 Deliberative Reasoning and Planning in LMs

Existing work on deliberative reasoning in LMs can be roughly categorized into three approaches: 1) Prompting-based approaches such as chain-of-thought (Wei et al., 2022b; Kojima et al., 2022) and plan-and-solve (Wang et al., 2023a) elicit reasoning and planning capabilities from pretrained LMs via advanced prompting methods but without any additional finetuning (Wei et al., 2022b; Kojima et al., 2022; Wang et al., 2022b;a; 2023a; Zheng et al., 2023b; Kim et al., 2023; Xue et al., 2023; Zhao et al., 2023; Sun et al., 2024; Yao et al., 2023; Hao et al., 2023). 2) Through a combination with external tools, LMs can learn to use e.g. calculators when appropriate (Schick et al., 2023; Chen et al., 2023; Paranjape et al., 2023) or receive feedback from domain specific solvers (Nye et al., 2021; Valmeekam et al., 2023). 3) Finally, finetuning on task-specific reasoning data can result in substantially improved in-domain performance for e.g. mathematical reasoning (Hendrycks et al., 2021; Cobbe et al., 2021; Havrilla et al., 2024), code generation (Shen et al., 2023), and navigation tasks (Carta et al., 2023). While the above three approaches lead to substantial performance gains on individual reasoning tasks, they deviate from the paradigm of large-scale self-supervised training on unlabeled data that gave rise to the dominance of LMs, which our method addresses. The Planning Tokens (Wang et al., 2023b) method is perhaps most relevant to our work: Here, the dataset is augmented with planning tokens that represent high-level information similar to our writing actions. However, planning tokens are generated by the LM itself rather than by an external planner, and are hence limited in their flexibility and ultimately performance. Most importantly, again their method is applied to a narrow mathematical reasoning dataset rather than a general-purpose corpus. Nonetheless, in our experiments, we compare our external planner to the LM-internal planning approach by Wang et al. (2023b).

### 2.2 Hierarchical Text Processing in Deep Learning

By planning in the space of abstract writing actions derived from text units (e.g., sentences), our method explicitly addresses the hierarchical nature of language. This is a common theme in NLP models; for example, hierarchical attention mechanisms have been employed for summarization (Nallapati et al., 2016), text classification (Yang et al., 2016) and machine translation (Miculicich et al., 2018). Several works have decomposed text generation tasks into sentence-level and word-level prediction tasks (Tan et al., 2017; Perez-Beltrachini et al., 2019; Marfurt & Henderson, 2021; Jhamtani & Berg-Kirkpatrick, 2020), but in contrast to our work do not condition on *abstract* writing actions. Ippolito et al. (2020) propose to do language modeling at the sentence-level, while we ultimately still perform language modeling at the token level.

### 2.3 Language Modeling

Language models have been an important subject of study for decades. While initial models were based on count-based n-gram LMs, the models that are ubiquitous today are based on

---

[1]The code is available on Github.

neural LMs (Bengio et al., 2000). Algorithmic advances are one major source of progress in neural language modeling (Ho et al., 2024), through e.g. recurrent LMs (Mikolov et al., 2010; 2011), specialized attention mechanisms such as pointer-networks (Vinyals et al., 2015; Merity et al., 2017), and eventually transformer-based (Vaswani et al., 2017) language models (Radford, 2018). While the contribution of our paper is on the algorithmic side, it is important to remember that, as Ho et al. (2024) identify, in recent years progress has been driven mostly by scaling up similar architectures (Radford et al., 2019; Brown et al., 2020; Hoffmann et al., 2022), which gives rise to the key motivation of our paper: New algorithms for planning and reasoning in LMs need to scale to unlabeled data to be relevant long-term.

**Related methods**  Among the important algorithmic improvements, Retrieval Augmented Generation (RAG) (Guu et al., 2020; Lewis et al., 2020) informs next-token generation by querying a large database of text. A special case of RAG, nearest neighbor language models (Khandelwal et al., 2020), *retrieve* the closest matches to the *past* based on similarity in a latent space, transform the distances into a probability distribution, and interpolate it with the LM's distribution *without any additional finetuning*. Orthogonal to the above, we propose a method that *predicts* a sentence into the *future* in order to inform the generation of the next tokens via finetuning from unlabeled data. In Controllable Text Generation (Zhang et al., 2024a; Mudgal et al., 2024; Guo et al., 2022; Deng & Raffel, 2023; Yang & Klein, 2021), generation is guided by user-provided attributes such as topic, style, or keywords. Our method can be seen as a form of controlled text generation, where guiding (latent) attributes are predicted by an external model rather than provided by the user. Aiming at separating the global context from the local context, topic-guided LMs (Lau et al., 2017; Wang et al., 2019) condition the generation on topic models inferred from the context. Recent results indicate that this does not improve language modeling because well-trained LMs already possess a good awareness of the current topic (Zheng et al., 2023a). In contrast, our planner predicts *future* writing actions, which often carry information that is structural rather than topical.

## 3   Methodology

We consider the general task of language modeling, i.e., estimating the probability $p(x_1 x_2 \ldots x_n)$ for any text sequence $X = x_1 \ldots x_n \in \mathcal{X}$. As is standard practice, we factorize $p(x_1 \ldots x_n) = \prod_{i=1}^{n} p(x_i | x_1 \ldots x_{i-1})$. However, in contrast to a standard LM, we condition our LM on additional writing actions $a \in \mathcal{A}$, which are predicted by an external planner module $P : \mathcal{X} \rightarrow \mathcal{A}$. To this end, we split the input $X$ into text units $X = t_1 \ldots t_m$ consisting of one or more tokens $t_j = x_1^j \ldots x_{n_j}^j$. A text unit could for example be a paragraph or chapter, or even be determined dynamically. However, in this paper we choose sentences for simplicity. We associate one writing action $a_j$ with each text unit $t_j$, which we write as $X' = a_1 t_1 \ldots a_m t_m.$[2] Hence, we frame language modeling as estimating $\prod_{j=1}^{m} \prod_{i=1}^{n_j} p(x_i^j | a_1 t_1 \ldots a_{j-1} t_{j-1} a_j x_1^j \ldots x_{i-1}^j)$, i.e., the probability of the next token in the current text unit given the sequence of previous text units with their corresponding writing actions and the previous tokens in the current text unit, via our language model $p_\theta$.

Our method consists of three core contributions as sketched in Figure 1: First, we show how to derive an abstract writing action for every text unit in our unlabeled data, providing us with training data. Second, we propose a simple planner for predicting the next writing action. Finally, we finetune the language model by conditioning on predicted actions.

### 3.1   Generating Abstract Writing Actions from Unlabeled Data

Given a training corpus $\mathcal{X}$ with articles $X = t_1 \ldots t_n \in \mathcal{X}$, we first embed every text unit into a low-dimensional vector $\mathbf{z}_j = \mathrm{E}(t_j) \in \mathbb{R}^d$ via a text encoder E (e.g. Sentence-

---

[2]We choose this notation for convenience. The actions $a_j$ are not in textual form.

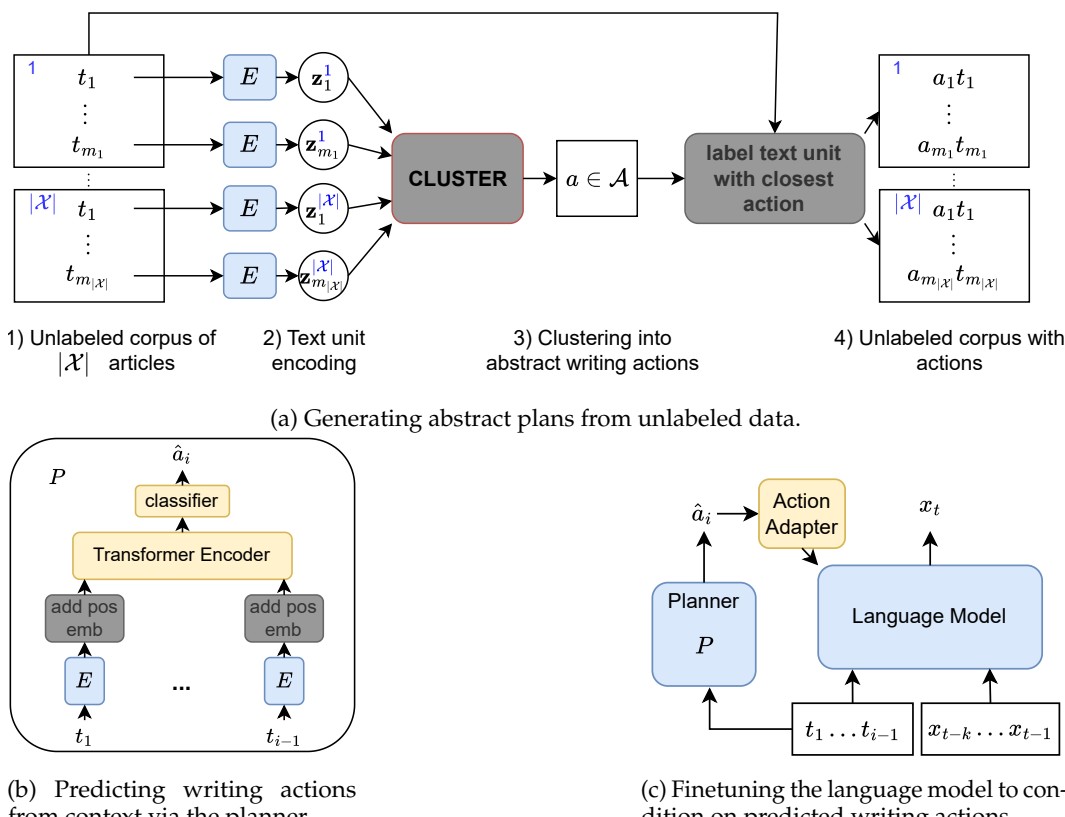

(a) Generating abstract plans from unlabeled data.

(b) Predicting writing actions from context via the planner

(c) Finetuning the language model to condition on predicted writing actions.

Figure 1: The three core phases of our proposed method to learn a planner from unlabeled data. Blue indicates frozen parameters, yellow indicates trainable parameters, and grey indicates no learnable parameters .

BERT (Reimers & Gurevych, 2019)). In principle, we could now associate each text unit with its corresponding embedding, i.e., choose the action embedding $\overrightarrow{a_j} = \mathbf{z_j}$. However, while this would retain most information, in this case $a_j$ is very difficult to predict due to the large number of options. Instead, we posit that it is necessary to compute abstractions that correspond to common writing patterns. Similar to Wang et al. (2023b), we employ a simple k-means clustering step in the embedding space to narrow down the possible actions. We reuse the resulting cluster centroids as the actions embeddings $\overrightarrow{a} \in \mathbb{R}^d$. Since the embeddings carry semantic information, we expect the resulting centroids to correspond to general writing patterns. We set $k = 1024$ by default. In Section 5.3, we investigate the effect of $k$ and what our clusterings have learned.

## 3.2 Predicting Future Writing Actions with a Planner Module

During inference, we do not have access to the true writing actions, so they have to be predicted. We could simply treat writing actions as tokens and try to predict them along with every other token by finetuning the language model on the new data, as is done in Wang et al. (2023b). However, we argue that it is beneficial to treat the prediction of writing actions differently from tokens, namely through an external prediction module, for three reasons: **(1) Importance:** Writing actions weigh higher in importance than individual tokens, as they carry information regarding entire text units (e.g., sentences, paragraphs), rather than subword units. **(2) Planning:** The predicted writing action has significant influence over the future generated text, demanding elaborate planning to ensure the right choice is made, which next-token prediction as in LLMs cannot provide. **(3) Modularity:** An external

planner is not specific to any language model. Hence, it can be trained once and henceforth be used for conditioning various language models downstream.

In this paper, we propose a simple planner that only predicts the next writing action. However, our framework can encompass planners of arbitrary complexity, allowing for long-term prediction multiple steps ahead and the integration of a look-ahead search, similar to successful reinforcement learning approaches to playing games (Schrittwieser et al., 2020).

We model action prediction as a classification task, by training it via cross-entropy to predict the next action obtained from unlabeled data in the first step (Section 3.1). The default choice for such a text classification setup is to embed the whole context at once. However, in preliminary experiments, we found it helpful to embed the input in the same space as the output. As illustrated in Figure 1b, our planner first embeds each text unit $t_j$ independently into a single vector $\mathbf{z}_j$ using the same (frozen) text encoder as in Section 3.1:

$$\mathbf{Z}_{i-1} := \{\mathrm{E}(t_1) + \mathbf{p}_1, \ldots, \mathrm{E}(t_{i-1}) + \mathbf{p}_{i-1}\}.$$

$\mathbf{p}$ is an absolute position embedding to integrate sequence information (Vaswani et al., 2017). Given the embedding matrix $\mathbf{Z}_{i-1} \in \mathcal{Z} = \left(\mathbb{R}^d\right)^+$ with $(\mathbf{Z}_{i-1})_j$ representing vector $\mathbf{z}_j$, we predict the subsequent writing action $a_i$. Given that the embedding is a set of vectors, we employ a simple Transformer encoder with mean pooling and a linear classifier head:

$$\mathbf{Z}'_{i-1} = \mathrm{Transformer}\left(\mathbf{Z}_{i-1}\right); \quad \mathbf{o}_{i-1} = \frac{1}{i-1} \sum_{j=1}^{i-1} \left(\mathbf{Z}'_{i-1}\right)_j; \quad \mathbf{pr}_\mathcal{A} = \mathrm{softmax}\left(\mathbf{W}_o \mathbf{o}_{i-1} + \mathbf{b}\right).$$

Here, $\mathbf{pr}_\mathcal{A}$ is the predicted probability distribution over actions. Since the semantic relationships between actions are encoded in their embeddings $\overrightarrow{a} \in \mathcal{R}^d$ which we obtained from clustering, we initialize the output matrix $\mathbf{W}_o \in \mathbb{R}^{|\mathcal{A}| \times d}$ with action embeddings.

### 3.3 Fine-tuning the Language Model Conditioned on Predicted Writing Actions

State-of-the-art LMs are already powerful next-token-predictors that possess various capabilities. When training these models to make use of the information in provided writing actions, we want to finetune them in a minimally invasive way in order to avoid catastrophic forgetting and overspecialization to a particular training domain. To achieve this, we propose an adapter-style conditioning module.

For an input sequence of length $P$, for each position $p \in P$, we select an action to be merged into the model. Specifically, we select the action whose centroid embedding is closest to the sentence embedding of the text unit to which the token at the *next* position $p+1$ corresponds. This is done because the final-layer prediction at that position $p$ predicts the probability of the token at position $p+1$. Call this action $a_{p+1}$. Following Zhang et al. (2023), we insert actions only in the last $L$ layers of the language model. In each layer $l$, we first embed the action using an embedding table $E_A^l$. We will show in Section 5.3 that it is beneficial to initialize $E_A^l$ with the action embeddings $\overrightarrow{a}_{p+1} \in \mathbb{R}^d$ that correspond to the centroids of the clustering phase. We project this embedding once more with a linear layer $\mathbf{W}_A^l \in \mathbb{R}^{d' \times d}$ to get the action representation $\mathbf{r}_p^l$ at position $p$ and layer $l$:

$$\mathbf{r}_p^l = \mathbf{W}_A^l E_A^l(a_{p+1}).$$

The action representation is then added element-wise to the $d'$-dimensional feature representation inside the transformer at layer $l$ and position $p$, just after the multiplication with attention weights, and just before the output projection. This is similar to the Llama-Adapter (Zhang et al., 2023) with adaptation prompt sequence length 1, except that we do not use gating, which we found not to be helpful. With action-information inserted into the model as explained above, we finetune the LM by minimizing cross-entropy for next-token-prediction as

$$\mathcal{L} = \mathbb{E}_{X=[t_1 \ldots t_{j-1} x_1^j \ldots x_i^j] \sim P(\mathcal{X})} \log p_\theta \left(x_i^j | \hat{a}_1 t_1 \ldots \hat{a}_{j-1} t_{j-1} \hat{a}_j x_1^j \ldots x_{i-1}^j\right),$$

where $P(\mathcal{X})$ is the distribution of sequences and $\hat{a}$ denotes planner-predicted writing actions.

An alternative approach is to finetune the language model with oracle writing actions, which we also evaluate in Section 5.1. One the one hand we expect that tuning with oracle actions will increase the mismatch between training (oracle actions) and testing (predicted actions), but on the other hand the more reliable oracle actions might induce the LM to rely more strongly on them.

## 4 Experimental Setup

### 4.1 Hypotheses

The goal of our paper is to demonstrate the viability of our method for learning an external planning algorithm from unlabeled data whose predictions inform language modeling. Hence, our experiments are designed to test the following hypotheses: **(H1)** Our method successfully integrates the information from predicted writing actions into language modeling. **(H2)** The external planner is more successful than an internal planner. **(H3)** Our method particularly improves performance in terms of text structure.

### 4.2 Experimental Details

In all our experiments, we train and evaluate our models on a subset of English Wikipedia articles. To this end, we utilize a cleaned Wikipedia dump from March of 2022 that is publicly available via Huggingface[3]. We train on a subset of 285310 articles (corresponding to roughly 300M tokens), we split off 1000 articles for early-stopping validation, and 1000 articles for testing. Due to the high computational cost of training a model, most of our experiments are performed on the small GPT2 model (Radford et al., 2019). However, in order to show that our method is viable for larger, more powerful models, we additionally perform an experiment with the recent OLMo 1B (Groeneveld et al., 2024). The full list of hyperparameters is provided in Appendix A.

In Section 5.2 we compare our external planner against the LM-internal planner by Wang et al. (2023b). Since this model uses finetuning to adapt the LM itself to predict writing actions, a fair comparison of internal and external planner requires adaptation of our method to this scenario, which we describe in detail in Appendix C. Briefly, the adaptation consists of inserting externally predicted writing actions as tokens, which allows every token that follows to condition on it via attention. Moreover, the adaptation allows finetuning of the original LM parameters.

### 4.3 Evaluation

**Primary evaluation**  The primary objective of our work is to improve language modeling whose standard metric is perplexity $PPL = \exp(\mathcal{L})$.

**Surface-level generation evaluation**  Because perplexity does not evaluate actual generations of the model, we also report ROUGE-2 (F1) (Lin, 2004) and MAUVE (Pillutla et al., 2021) scores. ROUGE-2 measures the overlap between generated text and real text in terms of bigrams. Concretely, given a real text $X = x_1 \ldots x_n = t_1 \ldots t_m$, we first use the trained models to generate text continuations $\hat{X} = t_1 \ldots t_i \hat{t}_{i+1} \ldots \hat{t}_{m'}$ given the prefix $t_1 \ldots t_i$ for some $i$. We then compare true and generated continuations at different lengths, we report the average of the different lengths in Section 5.1 and the individual results in Appendix B.

MAUVE is a metric designed to evaluate open-ended text generation. It aims to measure the divergence between the probability distribution of the model and the true data distribution. To do so, it takes a set of true articles and generated articles, and deduces a tractable probability distribution from each by embedding and clustering the articles to form histograms. It then calculates a divergence frontier (Djolonga et al., 2020) between the two distributions,

---

[3]https://huggingface.co/datasets/wikipedia. We use the "20220301.en" version.

which is analogous to a precision-recall curve applied to generative models. The area under the curve is the MAUVE score. Because MAUVE doesn't require pairwise comparisons, we generate texts unconditionally, i.e. without context. We generate 1024 tokens, which corresponds roughly to the average article length.

**Abstract generation evaluation**    Additionally, we hypothesize that our planning improves the language modeling performance in particular with respect to text structure. To this end, we measure how well model-generated texts corresponds to ground-truth texts in terms of writing action sequences. We use the procedure described in Section 3.1 to obtain the sequence of writing actions that corresponds to a real or generated text.

We provide two metrics for this: Levenshtein (Levenshtein et al., 1966) distance and latent perplexity (Deng et al., 2022). The Levenshtein distance (also referred to as edit distance) measures the number of insertions, deletions, and substitutions necessary to make the sequences equal. Similar to ROUGE-2, it compares generated text to ground-truth text in a pairwise manner. Hence, we report results for the same text continuations as for ROUGE-2.

For latent perplexity, we train a simple HMM-based critic to model the distribution of sequences of abstract writing actions of *ground-truth* text. Latent perplexity results as the perplexity of the sequence of abstract writing actions in the *generated* text. Latent perplexity evaluates open-ended text generation, and we thus report it on the same unconditionally generated texts as MAUVE.

**Planner evaluation**    While the end-goal of our work is improving language modeling, the way in which we aim to do that hinges on two components. First, how accurately the planner module can predict oracle writing actions, and second how successfully we can finetune the language model to make use of the information provided by the planner. To measure the former in isolation, we report two metrics: accuracy (i.e., percentage of writing actions predicted correctly), and average rank. The planner predicts a score for each possible action. To calculate the average rank, we order the predicted scores of each action from high to low and observe which rank the oracle action has on average.

### 4.4   Baselines

Since our method involves finetuning a language model via the action adapter (cmp. Figure 1c), improved performance over the language model with no finetuning (henceforth called **None**) is expected due to adaptation to the finetuning dataset (Wikipedia). However, since the goal of this paper is to identify how well the language model is able to make use of the information in the predicted writing actions (called **Predicted**), we would like to isolate the benefit introduced from predicted writing actions alone for our main experiments. To this end, we employ the **Fixed** baseline, which is finetuned on the same dataset, but always receives the same (uninformative) writing action regardless of the context. Finally, we employ the **Oracle** baseline, which always receives the true next writing action, to serve as an upper bound for planners trained on our extracted writing actions. The different baselines correspond to different writing actions being provided to the language model, but each baseline always uses the same type during training and testing. For **Predicted**, however, it is possible to train using either oracle actions (**Predicted-OA**) or planner-predicted actions (**Predicted-PA**).

Wang et al. (2023b) train an LM-internal planner, to which we compare our external planner. To enable a fair comparison that isolates the effect of internality/externality of the planner, we re-implement their approach in our framework. The implementation details can be found in Appendix C.

| Base-LM | Model | PPL ↓ | MAUVE ↑ | Latent PPL ↓ | ROUGE-2 ↑ | Edit ↓ |
|---|---|---|---|---|---|---|
| GPT-2 Small | Oracle | 23.92 | - | - | - | - |
| | Baselines | | | | | |
| | None | 32.89 | 0.20 | 438.10 | 0.013 | 4.26 |
| | Fixed | 26.69 | 0.38 | 352.8 | 0.015 | 3.88 |
| | Proposed | | | | | |
| | Predicted-OA | 26.94 | **0.45** | **91.6** | **0.019** | **3.69** |
| | Predicted-PA | **25.55** | 0.44 | 205.9 | 0.017 | 3.78 |
| OLMo 1B | Oracle/Oracle | 10.99 | - | - | - | - |
| | Baselines | | | | | |
| | None | 13.08 | 0.14 | 374.22 | 0.019 | 4.83 |
| | Fixed | 11.81 | 0.45 | 250.90 | 0.022 | 3.42 |
| | Proposed | | | | | |
| | Predicted-OA | 11.99 | 0.43 | **74.2** | **0.026** | 3.39 |
| | Predicted-PA | **11.46** | **0.56** | 178.2 | 0.025 | **3.31** |

Table 1: Model evaluation under different training and conditioning scenarios. Predicted-OA was trained with oracle actions, while Predicted-PA was trained with planner-predicted actions. The improvement of Predicted-OA over Fixed over all metrics indicates the extent to which the model learns to make use of the information in the action **(H1)**, and Oracle shows the best possible performance under perfect prediction of writing actions.

# 5 Results

## 5.1 Main Result

Table 1 shows the results. The main result is that the LM trained with planner-predicted actions (**Predicted-PA**) outperforms the **Fixed** baseline by 1.14 PPL for GPT-2 and 0.35 PPL for OLMo. It also outperforms **Fixed** on both surface-level (ROUGE-2, MAUVE) and abstract (Latent PPL, Edit) generation metrics. The results also show a trade-off between finetuning the LM on planner-predicted actions (**Predicted-PA**) and oracle actions (**Predicted-OA**). Finetuning on oracle actions damages perplexity, as it introduces a mismatch between training and testing data. Despite this, it is sometimes better in terms of generation metrics. An explanation can be found in the different extent to which the resulting LMs actually *follow the plans* proposed by their planner, i.e., whether they produce sentences that belong to the cluster corresponding to the action used in generating the sentence. We find that **Predicted-OA** follows the plan about 40% of the time, compared to only 20% for **Predicted-PA**. This makes sense, as oracle actions are more informative than predicted actions. Hence, there is a trade-off between reducing the train/test mismatch to achieve low perplexity and following the plan to achieve high generation quality. Finding a balance between these two is a promising direction for future work.

We also observe that there remain differences of 1.63 PPL and 0.47 PPL with respect to **Oracle**. While it is not possible to train a planner that is *exactly* equivalent to oracle access, this does suggest that a planner that is better at matching oracle actions can further improve language modeling performance. Moreover, in Appendix D we find that, while the oracle action is much better than a random choice, it is not optimal. Hence, further perplexity gains could be made by learning to consistently predict those actions that lead to an even lower perplexity than the oracle actions.

## 5.2 External vs. Internal Planner

Conducting a fair comparison between internal and external planner as described in Appendix C, we find that the external planner performs 1.22 PPL better than the internal planner by Wang et al. (2023b) if we restrict finetuning to parameter-efficient finetuning with LoRA, and .81 PPL better in full finetuning.

## 5.3 Ablations

**Model Architecture** In Table 2, we perform an ablation showing the contribution of several architecture components. Having a multi-vector representation improves the performance of the planner, which in turn is slightly beneficial when conditioning the language model on predicted actions. As expected, initializing the action embeddings with the respective cluster centroids is also crucial to the performance of the model. While initializing the planner with action embeddings helped for preliminary experiments at small scale, the effect disappears with more training data.

**Clustering** Figure 2 shows the model behavior as we increase the number of possible actions. The perplexity improves up to 8192 actions, after which it decreases again. This is explained by the tradeoff between informativeness and predictability: More clusters correspond to less compression of information contained in the sentence embeddings. On the other hand, more available actions make the prediction task harder, as indicated by an increasing average rank.

In Tables 6 and 7 in the Appendix, we analyze the ten largest clusters qualitatively. We find that they often correspond well to structures typically found in a Wikipedia article. For example, an article about a person typically contains information about a person's origin, education, and professional life in a certain order, which a planner would be able to predict in terms of writing actions. Some actions, however, are rather topical than structural. For example, cluster #7 contains mentions of Italian people and places. This indicates that our procedure of generating abstract writing actions (see Section 3.1) can be improved further, e.g. through an encoder whose representations contain more structural rather than topical information.

| Model | Acc. ↑ | Rank ↓ | PPL ↓ |
|---|---|---|---|
| full model | 24.7 | 36.4 | 25.55 |
| no sent rep | 21.9 | 38.3 | 25.59 |
| no P init | 24.3 | 37.3 | 25.51 |
| no AD init | 24.7 | 36.4 | 27.29 |

Table 2: Ablations. **no sent rep:** Instead of encoding sentences separately, encode whole context into one vector via mean-pooling. Use an MLP on top instead of a Transformer. **no P init:** Randomly initialize action output layer of the planner. **no AD init:** Randomly initialize action input layer of adapter.

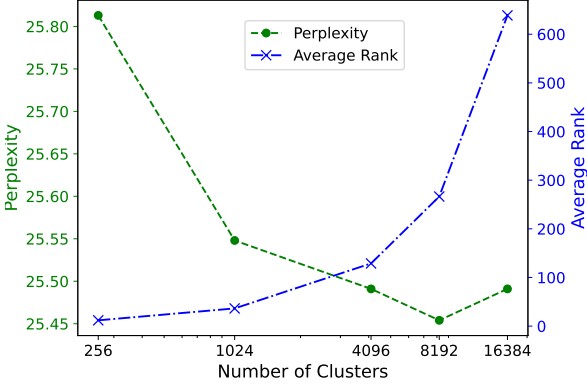

Figure 2: Performance by number of clusters.

| # | Content | Example sentence |
|---|---|---|
| 1 | Demographics | The rural population was 693,566 (89.03%) and urban 85,496 (10.97%). |
| 2 | Plot | During a private argument about whether or not they should move, Elaine reveals she is pregnant. |
| 3 | Business | 2010: Snecma and GE formed CFM Materials as a 50/50 joint vefnture. |
| 4 | Geography | Khuzama is situated in Jakhama circle of Kohima District in Nagaland. |
| 5 | Schools | The district consists of an elementary school, a middle school, and a high school. |
| 6 | Education | He has a bachelor's degree from the University of Florida. |
| 7 | Italy | Maffeo Olivieri (1484, Brescia — 1543 or 1544) was an Italian sculptor and wood carver. |
| 8 | Acronyms | Comput. |
| 9 | Origin | Shokler was born in Cincinnati on April 27, 1896. |
| 10 | Career | In 1954 he played an impressionable navigator opposite Gregory Peck in The Purple Plain. |

Table 3: Analysis of the ten largest clusters.

# 6   Discussion

The experimental results (Section 5.1) confirm that our method is able to enhance language model performance by integrating externally predicted writing actions **(H1)** inferred from unlabeled data. This finding holds both for small LMs and more powerful, larger LMs. Naturally, the effect is smaller for larger models because their general language modeling ability is already stronger.

As evidenced by experimental results in a fair comparison (Section 5.2), externality of the planner **(H2)** is beneficial. Besides performance, an external planner enables greater modularity such that the language model and the planner can be developed and reused independently.

The enhanced LM ability of our method can be attributed to a stronger ability to plan their writing structurally **(H3)**, since conditioning on predicted writing actions causes the model to follow the ground truth text structure, which is indicated by a lower Levenshtein distance and latent perplexity in comparison to finetuning without a planner (Section 5.1). Our qualitative analysis of clusters (Section 5.3) suggests that writing actions indeed correspond to common structures in the dataset.

Finally, our method's performance depends on the technical contributions that we have made, such as a better neural architecture for predicting writing actions (Section 5.3) and conditioning the LM on predicted actions rather than ground truth actions during training (Section 5.1).

## 6.1   Limitations

Our study is limited in some ways. First, while we try to show the generality of our method through experiments with a more powerful, medium-sized OLMo 1B model, due to computational constraints, we cannot train models of even larger scale and strength. Moreover, while we trained on the Wikipedia corpus, which is often considered general-purpose, its diversity is still considerably more limited than the datasets modern LLMs are trained with, which typically subsume a snapshot of the web. Finally, since LMs only achieve nontrivial performance on zero- and few-shot tasks after large amounts of pretraining compute, we do not evaluate the benefit of our method on such downstream tasks. In the future, we want to apply our method to an LLM of medium size (e.g. OLMo 7B) finetuned on a diverse dataset (e.g. Dolma (Soldaini et al., 2024)).

Second, while we improve over a standard architecture with a model better suited for predicting future text, there are several obvious avenues for improvement that remain unexplored. Since we obtain abstract writing actions via clustering in a text embedding space, they do not necessarily encode information that is best suited to help language modeling. We view learning actions tailored to this use case as a promising research direction. Moreover, we want to explore planning multiple text units in the future and using a search method such as beam search or Monte Carlo Tree Search at inference time to improve planning and find the action that optimizes downstream LM performance.

# 7   Conclusion

Large Language Models have impressive capabilities, but they lack the ability to plan their writing well. Our proposed method utilizes an independently trained, external planner module, whose outputs can be used to inform and improve the generation process of the language model. Crucially, the planner is trained from unlabeled data in a self-supervised fashion. These qualities enable researchers to develop and train new planners at large scale and publicly share them, allowing anyone to integrate new planning capabilities with the language model of their choice. Hence, we envision a future where progress on planning algorithms advances at a similar speed as language models today.

**Acknowledgments**

This research was financed by the CALCULUS project—Commonsense and Anticipation enriched Learning of Continuous representations—European Research Council Advanced Grant H2020-ERC-2017-ADG 788506, `http://calculus-project.eu/`.

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

## A Hyperparameters and Experimental Details

**Implementation**

The code is available on Github.

**Data processing**  All of our experiments are based on a cleaned Wikipedia dump from March of 2022 that is publicly available via Huggingface[4]. We do not apply any further preprocessing to the dataset.

For generating abstract writing actions (Section 3.1), we first use spaCy (Honnibal et al., 2020) to split every article into sentences. Then, we use MPNet-base-v2 (Song et al., 2020) via the SentenceTransformer library (Reimers & Gurevych, 2019)[5] to encode sentences into embeddings. For clustering, we use the existing k-means implementation from Scikit-Learn (Pedregosa et al., 2011) with k-means++ initialization (Arthur & Vassilvitskii, 2007) and run it for the default maximum of 300 iterations.

---

[4]`https://huggingface.co/datasets/wikipedia`
[5]`https://huggingface.co/sentence-transformers/all-mpnet-base-v2`

**Neural network implementation**  All methods compared in our paper are implemented within the same PyTorch-based (Paszke et al., 2019) framework. Language models are loaded via the Huggingface Transformers library (Wolf et al., 2020). Experiments with the GPT2 model[6] model were run on a single 12GB GPU, whereas experiments including OLMo 1B[7] were run on a single 24GB GPU.

**Evaluation**  To compute the Levenshtein distance, we use the python-Levenshtein package (`https://pypi.org/project/python-Levenshtein/`) with default settings. For ROUGE and MAUVE, we used the python packages 'rouge' and 'mauve-text', respectively. For latent perplexity, we made minor changes to the implementation by Deng et al. (2022), available at `https://github.com/florianmai/criticize_text_generation`.

**Hyperparameters**

Table 4 shows hyperparameters used for our experiments.

| Hyperparameter | Value |
|---|---|
| Context window size | 128 |
| Train \| test \| val split sizes | 285310 \| 1000 \| 1000 |
| K-means initialization | k-means++ |
| Number of tokens generated for edit distance | 128 |
| Default action count | 1024 |
| Action embedding dimension | 768 |
| **Language Model Finetuning** | |
| Batch size | 32 |
| Learning rate (lr) | 1e-4 |
| **Planner Training** | |
| Batch size | 32 |
| Learning rate (lr) | 1e-4 |

Table 4: Hyperparameter Settings

## B Detailed ROUGE and Edit Distance Results

Table 5 shows the Rouge-2 and edit distances between true and generated continuations, when cut off at different generation lengths. Because the edit distance scales linearly with the number of tokens, we normalize results: for 128 tokens we report the actual edit distance, for 256 the edit distance divided by 2, etc.

## C Internal vs External Planner Setup

We want to compare our approach that uses an external planner to using the language model itself as a planner, as is done in Wang et al. (2023b).

If $V$ is the vocabulary size, $W$ the number of writing actions, and $h$ the hidden dimension, using the language model as planner entails expanding the $V \times h$-dimensional final prediction layer to a $(V + W) \times h$-dimensional layer. Just training those extra $W \times h$ parameters does not suffice to let the language model learn to predict planning tokens, which is why Wang et al. (2023b) also finetune the existing language model parameters (either full-finetuning, or parameter-efficient finetuning with LoRA (Hu et al., 2022)).

---

[6] `https://huggingface.co/openai-community/gpt2`
[7] `https://huggingface.co/allenai/OLMo-1B`

| Base-LM | Model | 128 | 256 | 512 | 1024 | 2048 |
|---------|-------|-----|-----|-----|------|------|
| | | ROUGE-2 ↑ | | | | |
| | None | 0.0125 | 0.0134 | 0.0136 | 0.013 | 0.0116 |
| | Fixed | 0.0148 | 0.0167 | 0.0165 | 0.0154 | 0.01 |
| | Predicted-OA | **0.0185** | **0.0205** | **0.0211** | **0.0196** | **0.02** |
| GPT-2 | Predicted-PA | 0.0166 | 0.0178 | 0.0184 | 0.0169 | 0.01 |
| | | Edit ↓ | | | | |
| | None | 4.82 | 4.42 | 4.12 | 4.01 | 3.94 |
| | Fixed | 4.63 | 4.09 | 3.76 | 3.50 | 3.39 |
| | Predicted-OA | **4.39** | **3.88** | **3.55** | **3.35** | **3.25** |
| | Predicted-PA | 4.45 | 3.92 | 3.67 | 3.50 | 3.37 |
| | | ROUGE-2 ↑ | | | | |
| | None | 0.0212 | 0.021 | 0.0194 | 0.0168 | 0.0141 |
| | Fixed | 0.0242 | 0.0247 | 0.023 | 0.0203 | 0.0165 |
| | Predicted-OA | 0.0275 | **0.0296** | **0.0286** | **0.0255** | **0.0209** |
| OLMo | Predicted-PA | **0.0277** | 0.0284 | 0.0273 | 0.024 | 0.0192 |
| | | Edit ↓ | | | | |
| | None | 5.01 | 4.68 | 4.71 | 4.84 | 4.92 |
| | Fixed | 4.21 | 3.69 | 3.38 | 3.07 | 2.77 |
| | Predicted-OA | **4.15** | **3.64** | 3.29 | 3.06 | 2.84 |
| | Predicted-PA | 4.2 | **3.64** | **3.25** | **2.89** | **2.55** |

Table 5: Details of ROUGE-2 and Edit distance results for different numbers of tokens.

Besides having an external planner, our model also differs from Wang et al. (2023b) in the way predicted actions are presented to the model. We present the predicted writing actions to the language model with the adapter-style conditioning module detailed in section 3.3, whereas Wang et al. (2023b) insert the actions like tokens in between sentences. The insert-style requires extending the *input* embedding matrix of the language model in the same way as the final prediction layer above.

Consider a sequence of tokens $X = x_1^1...x_{n_1}^1 x_1^2...x_{n_2}^2......x_1^m...x_{n_m}^m$, where $x_i^j$ is the $i$th token of the $j$th text unit in the sequence. Let $A = a_1...a_m$ be the corresponding sequence of writing actions where each $a_i \in \{0...W-1\}$. Call idx-$x_i^j$ the index in the vocabulary of $x_i^j$, we then pass the following indices to the language model:

$$(a_1 + V) \text{ idx-}x_1^1 \ldots \text{idx-}x_{n_1}^1 \ldots \ldots (a_m + V) \text{ idx-}x_1^m \ldots \text{idx-}x_{n_m}^m$$

To isolate the effect of using an external planner vs using the language model as planner, we evaluate our model in an adapted setting where we also fine-tune original parameters, and condition on planner-predicted actions in insert-style rather than adapter-style.

Because learning to plan and learning to condition on the plan are intertwined for the internal planner, it does not allow training just a planner separately beforehand in order to condition on planner-predicted actions. Hence we condition the internal planner on oracle actions during training.

When training the model to condition on oracle actions, there is a slight mismatch with how it will be used at generation time (when it uses self-predicted actions): During training with oracle actions, the action embedding always corresponded to the centroid closest to the embedding of the text unit following it, while that does not necessary hold during generation. One option to reduce this mismatch during generation would be to retroactively change past predicted actions in the context of the language model to match the generated sentence that follows them. We leave evaluation of this option for future work.

## D  Are Oracle Actions Optimal?

An oracle action is the action whose clustered embedding is closest to the concrete embedding of the true next sentence. Table 1 showed that conditioning on an oracle code consistently leads to lower perplexity than conditioning on a fixed or predicted code. A natural question is whether providing a language model with the oracle action leads to *lowest* perplexity among all actions.

To test this, we first evaluate every action at every step, resulting in a ranked list of actions per step. Then, we compute the perplexity that one would get when always choosing the $k$-th best action. We plot the the results for every $k$ in Figure 3.

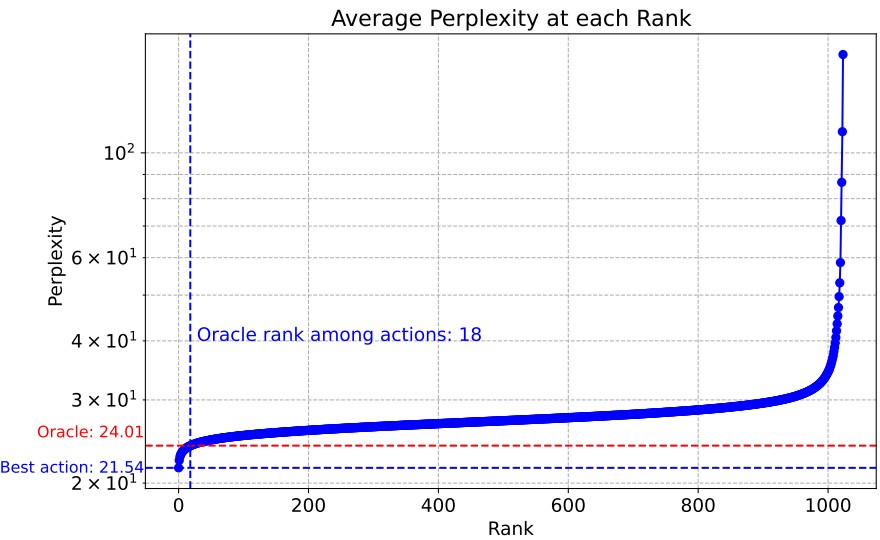

Figure 3: The blue dots show what the average perplexity is when conditioning on the $k$'th best action (in terms of what perplexity it leads to), with the rank $k$ on the horizontal axis. The red horizontal line displays the average perplexity of selecting the oracle code, the blue vertical line shows the rank with the nearest average perplexity to the oracle perplexity.[8]

The figure shows that the average perplexity using oracle codes is equivalent to using the 18th best action (out of 1024) in each step, meaning that on average there are 17 actions that would lead to a lower perplexity than the oracle action.

The fact that better-than-oracle-actions exist does not guarantee that it is feasible to train a planner that is able to consistently find those better-than-oracle-actions. One possible alternative explanation is the better-than-oracle actions are better purely due to luck: If one would make 1024 noisy variations of the oracle action embedding, some of those would also be better than the oracle embedding by chance. We investigate this alternative hypothesis by adding Gaussian noise ($\mu = 0$, $\sigma$ set to empirical standard deviation across the action embeddings over all layers) to the oracle action embeddings.

Figure 4 shows that indeed about half of the noise variations outperform the oracle embedding. However, it also shows that the lowest perplexity achievable by selecting the best random noise is significantly higher than that of selecting the best alternative action. This suggest random luck is not the full explanation for the success of the better-than-oracle actions, and it might be feasible to predict them consistently.

If indeed it is feasible, this suggest the planner can be made more effective by training it to predict the value of an action directly (in terms of the perplexity it leads to), rather than only training it to match oracle actions. A planner that can successfully estimate the values of its actions could then be combined with an algorithm like MCTS (Browne et al., 2012) at

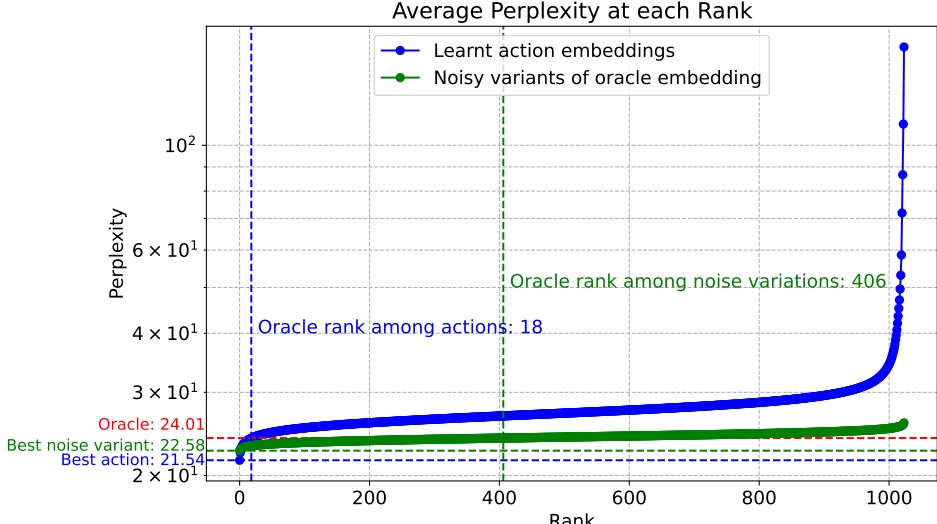

Figure 4: The green curve shows what the average perplexity is when conditioning on the $k'$th best noisy variation of the oracle action embedding (in terms of what perplexity it leads to), with the rank $k$ on the horizontal axis. The green dotted horizontal line indicates the perplexity of the best noise variant, and the green dotted vertical line the rank with the nearest average perplexity to the oracle perplexity among noise variations. The rest is the same as in Figure 3.

inference time for even greater performance gains. We consider this a promising avenue for future work.

## E   Examples of Clusters

Table 7 shows an extension of Table 3, i.e., multiple examples per action cluster.

## F   Complexity Analysis

In the following, we discuss the number of parameters that our method requires as well as the computational complexity.

**Parameter count**   The base language model has approximately 130M (GPT-2) or approximately 1B (OLMo-1B) parameters. The adapter needed for finetuning add about 14M parameters. The planner consists of a sentence transformer with 110M frozen parameters plus a small Transformer encoder with 6M additional parameters. Having a separate planner with its own sentence representations means we can reuse the same planner for different language models. As a memory-efficient alternative we can use the sentence representations of the LM (which have slightly worse overall performance), losing the portability advantage.

**Model complexity**   The complexity (number of computations) of our proposed sentence-based planner (Section 3.2), is $\mathcal{O}(\frac{N^2}{M})$ for the representation function (computation of $\mathbf{Z}'_{i-1}$) plus $\mathcal{O}(M^3)$ for the prediction function, where $N$ is the total number of tokens and $M$ the number of sentences. Since the representation function (110M parameters vs 6M for the prediction function) represents the vast majority of the complexity, invoking the planner is

---

[8]Because evaluating perplexity conditioned on all codes is slow, the average is taken only over 58 articles for this graph.

| # | Content | Example sentences |
|---|---------|-------------------|
| 1 | Demographics | **1:** According to the 2011 census, it had 841 inhabitants, all of whom were Albanian. **2:** The population was 23 as of 2002. **3:** Based on the 2007 national census conducted by the Central Statistical Agency of Ethiopia (CSA), this woreda has a total population of 112,396 of whom 56,245 are men and 56,151 women, no urban inhabitants were reported. **4:** 1790 - Population: 16,014. **5:** Its population as of 1990 was 344; its population as of 2000 was 680. |
| 2 | Plot | **1:** Feeling angry and betrayed, Nadine separates from Chloe, but they make amends. **2:** The pair ultimately decide to move in together after Britney kicks out her previous roommate. **3:** After Coré's death, Shane becomes strange and distant. **4:** When Nico finds out that he never showed the police Antonia's diary she questions their relationship and Philipp's love for her again. **5:** Louise reveals that she has been helping both Bob and Gene with their pranks without the other knowing and announces that she is quitting. |
| 3 | Business | **1:** Then in May 1998, all of PolyGram and its associated labels were purchased by Seagram which announced its plan to integrate Poly-Gram with UMG to produce an estimated cost savings, within a couple of years, of between US$275 million and $300 million annually. **2:** As of April 2020, the ownership of the pipeline was as represented in the table below. **3:** Wendy's International is owned by The Wendy's Company. **4:** October 2007: Nakheel announced the sale of Ireland, and Shanghai in October 2007. **5:** In 2016, Endeavor – a Hollywood-based entertainment group – acquired a reported 70%-controlling stake in Frieze, which includes its publishing, art fair and music interests. |
| 4 | Geography | **1:** Districts of Khyber Pakhtunkhwa **2:** and in Kyrenia District. **3:** Birhors are found mainly in the area covered by the old Hazaribagh, Ranchi and Singhbhum districts before these were broken down into numerous smaller units, in Jharkhand. **4:** Perumpanachy falls under Madapally Panchayat and Chanaganacherry Thaluk. **5:** It is about from Kuknur and from Lakkundi. |
| 5 | Schools | **1:** The old high school building is now being reused as the middle school. **2:** 2 High School is one of the four-star high schools in Jiangsu Province. **3:** The school has complete grade school levels, and high school levels. **4:** The sole entrance and exit to the school is along Texas State Highway Loop 1604. **5:** Students that attend the school come from the central parts of Prince County. |

Table 6: Analysis of the ten largest clusters. (Part 1)

roughly $M$ times less expensive than the language model (GPT-2: 128M parameters, OLMo: 1B parameters), which have complexity $\mathcal{O}(N^2)$.

At training time, due to its externality, the planner can be trained once and henceforth be integrated with a variety of language models. Therefore we focus on inference-time cost here. The planner consists of a representation function (sentence-transformer) and a prediction function (Transformer encoder on top of sentence embeddings).

*Representation function*: The bulk of the compute of the planner is in the representation function (110M parameters). Since our architecture encodes sentences in the previous context independently, sentence embeddings can be efficiently cached to avoid re-encoding when moving from one sentence to another. Computing the sentence embedding with a Transformer-based architecture has quadratic complexity in the length of the sentence. Assuming a simplified view where each sentence has length $N/M$, where $N$ is the total number of tokens in an article and $M$ is the number of sentences, encoding the necessary representations for all calls to the planner in an article has complexity $\mathcal{O}(M \cdot (N/M)^2) =$

| # | Content | Example sentences |
|---|---------|-------------------|
| 6 | Education | **1:** In 1957, he graduated with a thesis on Henry James. **2:** In 1996, he graduated with a BA in Political Science and Government, Philosophy, and Business Administration. **3:** He studied at Dalton School in New York and The Putney School in Vermont, later on graduating from Harvard University in 1965, in Far Eastern history. **4:** Guinle meanwhile earned a degree in Accountancy from the University of Morón in 2000. **5:** In 1986, Darren Barrett attended Berklee College of Music on a full scholarship, receiving a BA in Professional Music in 1990. |
| 7 | Italy | **1:** Non qui, Barbara, nessuno ci sta guardando (1989) **2:** Marchesa Olga Benucci Granieri Solaro — Monaldo wife, mother of Vittoria and Costanza and mother-in-law of Martino, Olga is a very caring woman to the image of his own family. **3:** Ugo Gregoretti (28 September 1930 – 5 July 2019) was an Italian film, television and stage director, actor, screenwriter, author and television host. **4:** This was the second Garibaldi group that succeeded the first group of the same name. **5:** Gaetano Marco "Guy" Nardulli (born May 31, 1974 in Norridge, Illinois) is an American actor and producer who is most associated for his character role as a street fighter turned MMA professional fighter in the 2006 movie -And Then It Breaks with actress Anne Dudek. |
| 8 | Acronyms | **1:** Soc. **2:** Dir. **3:** Ber. **4:** Ed. **5:** 2nd ed. |
| 9 | Origin | **1:** Ashley Sessa was born and raised in Schwenksville, Pennsylvania. **2:** Born on August 9, 1883, in Reading, Pennsylvania, Daisy Elizabeth Adams was educated in Reading, Pennsylvania. **3:** He was born in Keokuk, Iowa in 1877. **4:** Ada E. Schnitzer was born November 22, 1887, in Hannibal, Missouri, the daughter of O. C. Schnitzer. **5:** Morse was born in Abington, Massachusetts on August 19, 1911. |
| 10 | Career | **1:** He appeared in the 1995 Seinfeld episode "The Beard" playing Robert's boss. **2:** Tisch also made an appearance on the reality show Shark Tank in season 5. **3:** He starred as the surgeon Theodor Hristea, who, after some of his belongings are stolen, involves himself in the inquiry and directs the interrogation of a seemingly innocent man. **4:** He also appeared, uncredited, in the movie It's a Wonderful Life (1946), playing piano in the scene where George Bailey gets thrown out of Nick's Bar. **5:** Clint Eastwood was among the first of many actors to adopt this wandering ronin with no name persona for foreign films, which he used to great effect in his Western roles, especially in Spaghetti Westerns directed by Sergio Leone where he played the Man with No Name, a character similar to Mifune's seemingly-nameless ronin in Yojimbo. |

Table 7: Analysis of the ten largest clusters. (Part 2)

$\mathcal{O}(\frac{N^2}{M})$. In contrast, a non-sentence-based representation (see ablation 'no sent rep' in Table 2) has complexity $\mathcal{O}(N^2)$. Since there are many sentences in an article, the sentence-based planner is considerably more efficient.

*Prediction function*: The additional Transformer encoder is lightweight (6M parameters) and has complexity $\mathcal{O}(M'^2)$, where M' denotes the number of sentences in the current context. Overall, this brings the complexity to $\mathcal{O}(M^3)$ across the article. This could potentially be brought down to $\mathcal{O}(M^2)$ by using a Transformer decoder instead of a Transformer encoder for the prediction function, which allows for caching due to causal masking.

# G Color-coded Example Article

In order to give the reader an impression of how abstract writing actions are distributed within an article, below we print an article that is color-coded with different actions. Different text colors correspond to different actions.

## G.1 Article

In 1917, Weeks defeated incumbent John J. Mullen by 230 votes to become Mayor of Everett. The Boston Daily Globe described the race between Mullen and Weeks as "one of the bitterest campaigns in years" and in his inaugural address, Weeks referred to his predecessor as a "caterwauling demagogue" and vowed to overturn many of his acts, including firing of Police Chief William E. Hill and the closure of the Everett Tuberculous Hospital. In 1918, Christopher Harrison defeated Weeks by 390 votes, with Mullen, who supported Harrison after being eliminated in the preliminary election, taking credit for "putting [him] over".

In 1922, Weeks was the Progressive Party candidate for United States Senate. He finished sixth with less than 1% of the vote. In 1923 Weeks moved to Reading, Massachusetts. However, in 1933, he returned to Everett to run for Mayor. He made the runoff election, but was defeated by another former Mayor, James A. Roche. During the 1934 gubenatoral election Weeks supported Democrat James Michael Curley. In 1935 Curley appointed Weeks to the State Alcoholic Beverage Control Commission. In 1941, Weeks again ran for Mayor of Everett. He finished last in a four candidate primary.

Legal career In 1922, Weeks defended George H. Mansfield, a former Everett resident who was charged with murdering his lover, Alice Jones. The medical examiner later ruled that Jones committed suicide and District Attorney Thomas C. O'Brien asked the grand jury to return no bill against Mansfield. In 1924, Weeks served as a special counsel for defendants accused of being part of a extortion ring led by former Middlesex County district attorney William J. Corcoran. They were found guilty and Corcoran was sentenced to 7 to 10 years in prison. Weeks also defended Corcoran when he and Daniel H. Coakley were also charged with conspiracy to extort later that year. They were found not guilty on all counts. In 1927 Weeks represented Jerry Gedzium, a convicted murder who was appealing his death sentence. The conviction was upheld by the Massachusetts Supreme Judicial Court.

See also 1916 Massachusetts legislature

References

1880 births 1972 deaths Mayors of Everett, Massachusetts Massachusetts lawyers Massachusetts Progressives (1912) Massachusetts Republicans Members of the Massachusetts House of Representatives People from Reading, Massachusetts

