# OpenReview forum: "Learning to Plan for Language Modeling from Unlabeled Data"
_colmweb.org/COLM/2024/Conference — COLM_

### Official Review · Reviewer_Btei · 2024-05-14

**Rating:** 7
**Confidence:** 3
**Ethics Flag:** 1

**Summary:**

The paper proposes to improve language models by including a planner that predicts the next writing action, in addition to the tokens. The approach is to first encode all the sentences of the model, then cluster them into writing actions, use a planner to predict the next action and fine-tune the LM to condition on the predicted planning actions.

===

Update after rebuttal:

After reading all of the other reviews and the authors' responses, I am updating my score from 6 to 7.
One thing that could be improved, is phrasing in the beginning of the paper what exactly are actions and what planning them means. At first, it is not very clear and I had the impression that you were talking about edit actions (later it was specified, but it would be good to clarify this in the beginning).

**Questions To Authors:**

- What does "internal planner" mean in this context - is this fine-tuning of the model?
- It could be interesting to discuss the complexity of the model with the added planner - does the model perform much slower, and what is the trade off between memory requirements and the improvement of the language model?

**Reasons To Accept:**

The proposed approach of using an action planner is meaningful, and seems to improve the model performance.

**Reasons To Reject:**

From the beginning of the paper, it is not very clear what the writing actions are, which makes it a bit hard to understand what exactly is the idea and what the model is trying to learn.
The evaluation is a bit hard to follow - particularly the separation of Predicted/Fixed/Oracle.

---

> ### Author Rebuttal · Authors · 2024-05-31
>
> *Reasons to Reject*:
> 1. **Definition of writing actions**: Abstract writing actions are equal to the indices of the clusters obtained in the clustering step, which we already clarify in the third paragraph of the introduction and in Figure 1. Intuitively, they correspond to common writing patterns in the dataset, e.g. origin or profession in a biography. We will add further clarification to the abstract.
> 2. **Evaluation**:  We will add a sentence to Section 4.4 and the caption of Table 1 clarifying that Predicted/Fixed/Oracle actions can be evaluated both at training and test time. This evaluation setup allows for valuable insights such as that training the LM with oracle actions rather than predicted actions leads to worse performance due to the exposure bias.
>
> *Questions to Authors*:
> 1. **Internal planner**: As we refer in Section 4.4. the details of the internal planner (which corresponds to [1] which we reimplement as a baseline) are described in Appendix B. Intuitively, the language model acts as an internal planner that treats writing actions just like every other token. Those tokens are added to the vocabulary and the LM is finetuned on action-inserted text.
> 2. **Model complexity**: The complexity (number of computations) of our proposed sentence-based planner (Section 3.2) is $\mathcal{O}( \frac{N^2}{M} )$ for the representation function (computation of $ Z_{i-1} $, assuming caching of sentence embeddings during inference) plus $\mathcal{O}(M^3)$ for the prediction function (computation of $\bf{Z’}_{i-1}$), where $N$ is the total number of tokens and $M$ the number of sentences, where we consider a simplified analysis where each sentence has length N / M. Since the representation function (110M parameters vs 6M for the prediction function) represents the vast majority of the complexity, invoking the planner is roughly $M$ times less expensive than the language model (GPT-2: 128M parameters, OLMo: 1B parameters), which has complexity $\mathcal{O}(N^2)$.
> We will add a more elaborate discussion to the appendix of the paper.
>
> [1]  Xinyi Wang, Lucas Caccia, Oleksiy Ostapenko, Xingdi Yuan, and Alessandro Sordoni. Guiding Language Model Reasoning with Planning Tokens, December 2023b. URL http://arxiv.org/abs/2310.05707. arXiv:2310.05707 [cs].

---

> > ### Author Response · Authors · 2024-06-07
> > **Thank you for engaging with our rebuttal!**
> >
> > Given the extra page in the camera-ready version, we will add clearer definitions of actions and planning with them early in the paper.

---

### Official Review · Reviewer_Dmpy · 2024-05-14

**Rating:** 7
**Confidence:** 4
**Ethics Flag:** 1

**Summary:**

This work proposed a hierarchical way of generating text, where a high-level planning action is generated before generating the tokens in a sentence. These actions are induced by clustering sentence embeddings into K clusters, and are used to train two models: an action prediction model which predicts the action of the next sentence from the current sentence using a transformer; and a pretrained language model which is finetuned by injection actions into it. Experiments are conducted on language modeling, and the main evaluation happens at two levels: at a surface level, this method achieves a lower PPL compared to baselines; and a high-level evaluation of actions in a conditional setting, where the actions corresponding to the predicted continuation are closer to the ground truth actions compared to baselines.

===Post Rebuttal Update===
After reading authors' rebuttal, my concerns about surface-level and high-level generation evaluation have been resolved (Weakness 2 and 3). Besides, authors have clarified how the model knows sentence boundaries at generation time as well as the number of introduced parameters (which seems small). Overall, I like the idea of separating high-level planning with low-level realization, and would love to see followup works on pretraining a generally useful high-level writing planner. Therefore, I have bumped up my rating from 6 to 7.

**Questions To Authors:**

1. According to table 2, isn't no P init better in terms of PPL and edit distance?
2.  It seems that the action prediction doesn't take into account previous actions but instead only previous text, is the rationale that previous text contains information about previous actions?
3. What's the intuition that finetuning using oracle actions is worse than using predicted actions? Is it some sort of exposure bias argument?
4. Can you visualize an example article by color-coding actions in a full article? I think that would be more useful than table 6.
5. I think this statement "However, there remain differences of 1.63 PPL and 0.47 PPL with respect to Oracle ( Oracle/Oracle ), indicating that less than half of the potential of integrating the information in the writing action is actuated due to a suboptimal planner" doesn't consider the fact that the oracle actions might leak information about the text, so the PPL it achieves might not be achievable. For example, if you use a very large action space, then each action might already be able to recover every word in the sentence, and the PPL (at least for training) would be very low.

**Reasons To Accept:**

1. It is an interesting idea to separate high-level planning with low-level surface realization.
2. Experiments showed improved results both at low-level realization and high-level planning.

**Reasons To Reject:**

1. Some details are not clear. For example, the actions are at a sentence level, but it is not clear from the paper how the model knows when to transition to the next action at inference time: is it by introducing a special end-of-sentence token? Or by using some punctuations like "." or "?" to mark sentence boundaries?
2. The high-level action evaluation using the edit distance between the actions of ground truth continuation and generated continuation does not consider that there might be multiple possible ways to continue writing an article (even at the high level). It would be nice to see results using better evaluations, such as comparing the distribution of generated action sequences versus the original action sequences, as in "Model Criticism for Long-Form Text Generation" (https://aclanthology.org/2022.emnlp-main.815/).
3. For surface-level evaluation, this work mainly relies on PPL, but not on metrics that directly evaluate the generations themselves. It would be nice to see evaluation results under open-ended generation metrics such as MAUVE (https://openreview.net/pdf?id=Tqx7nJp7PR).
4. The method introduces additional parameters (the sentence BERT, and the transformer used in planner) compared to the baseline language model. I think it would be nice if the authors show a parameter count comparison.

---

> ### Author Rebuttal · Authors · 2024-05-31
>
> *Reasons to reject:*
> 1. We use a simple unigram classifier for **detecting sentence boundaries**, which classifies a token to be at the end of a sentence if it appears more often at the end of a sentence in our training dataset than not, which reaches 99.3% accuracy and 88% f1-score, which is invoked during generation.
> 2. & 3. We now implemented **latent PPL** (lower=better) with abstract writing actions as latent variables. We trained the same HSMM critic as used for the discourse coherence experiment in the paper you provided, which achieves a test latent PPL of 100.8. Furthermore, we provide **MAUVE** scores (higher=better).
>
> | |GPT-2 Ours|GPT-2 Fixed|OLMo Ours|OLMo Fixed|
> |---|---|---|---|---|
> |Latent PPL|204.9|352.3|168.95|249.74|
> |MAUVE|.429|.379|.555|.364|
>
> 4. **Parameter count**: The base language model has ~130M (GPT-2) or ~1B (OLMo) parameters. The adapter parameters are about 14M parameters. The planner consists of a sentence transformer with 110M frozen parameters plus a small Transformer encoder with 6M additional parameters.
> Having a separate planner with its own sentence representations means we can reuse the same planner for different language models. As a memory-efficient alternative one can reuse the sentence representations of the LM, losing the portability.
>
> We will add all new results and extra information to the final paper.
>
> *Questions to Author:*
> 1. We address the **impact of P** init in Section 5.3 “Model Architecture”, where we state that while it was beneficial in our preliminary experiments on little data, it doesn’t help with more training data.
> 2. Indeed, since the previous actions are derivable from the previous text, **adding the actions explicitly does not add any new information**. We suspect adding it might help when training on little data, but would not be beneficial with more data.
> 3.  Exposure bias is indeed **the reason for training the LM on predicted actions**, as we discuss in Section 5.1 in the paper (“it decreases the train/test mismatch”).
> 4. **Color-coding actions** is a nice visualization. We will add it to the final paper.
> 5. You are correct that **Oracle/Oracle performance is likely not achievable** since there is no such thing as a perfect planner. Note, however, that the level of leakage of concrete text is small due to the level of abstraction of the actions (millions of sentence embeddings clustered into 1024 actions). We will modify this statement accordingly.

---

> > ### Author Response · Authors · 2024-06-07
> > **Thank you for engaging with our rebuttal!**
> >
> > Thank you for your encouraging words. We look forward to continuing to work on pretraining generally useful writing planners.

---

### Official Review · Reviewer_BMkB · 2024-05-23

**Rating:** 6
**Confidence:** 4
**Ethics Flag:** 1

**Summary:**

This paper presents a method for explicit planning for LLMs for text generation:
1. It breaks text generation into “chunks” (or sentences”); and for generating the next chunk, it explicitly injects an “action” variable, obtained from the planner module,  to condition the next chunk generation.
2. It trains a separate transformer model to predict a sequence of action variables, as the “plan” of the generation.
3. The action variable is obtained by clustering the sentence level embeddings for sentences in the training corpus, which means no explicit supervision.

Empirically, the authors find that the planning can somewhat improve generation quality (in terms of lower perplexity).

**Questions To Authors:**

There are some missing references:
FUDGE: Controlled Text Generation With Future Discriminators
For an in depth analysis of controlled generations, table 1 in Probabilistic Inference in Language Models via Twisted Sequential Monte Carlo could be a helpful pointer.

**Reasons To Accept:**

This paper presents an interesting idea that explicitly models the “planning” of the writing then uses it to guide/improve the generation. It automatically derives the planning in an unsupervised fashion using sentence-level embeddings of the training data. It separates the planning module and the language model, and the needed training is relatively lightweight. This idea is relevant to some of the previous work like sentence-level language modeling (Ippolito et al) and FUDGE (Yang and Klein).

**Reasons To Reject:**

This paper can be improved as follows:

Methodology:
1. The reference of “plan” / “action” is based on the sentence embeddings rather than explicit writing plans like “write a sentence about concept X”, “detail the characteristics of the city”. In essence, the planning module is doing sentence level language modeling. I would recommend the authors clarify that in the paper and I would not call the $a$ variables as “actions” but just next sentence embeddings.
2. Because the author uses clusters of semantic embeddings, the granularity for expressing the fine-grained actions for the next sentence might not be fine-grained enough to guide the next generation. For example, in table 3 and table 6, the authors show examples of the major clusters of the action vectors – it only reveals major semantic/topic differences like business/geograph. That fails to capture the fine-grained control of the details during the writing process – e.g., two consecutive sentences might talk about the same general topic but focus on different aspects/details. Hence, the usefulness of the planning module is actually unclear (which is also reflected in the empirical results).
3. While it’s good to decouple the planning module from the language models, currently the training of the planning module does not use any signal from the language models, which might make the plans less helpful. Alternatively, one can use sentence representation from the same language model to train the planning module (rather than sentence bert).

Experimental design and results:
1. Missing baselines: In table 1, the author only compares with a baseline that does not train the language model (the None column). At least the baseline where simply fine-tuning the language model on the same training data using a regular causal LM objective should be compared.
2. Perplexity might not be the best measurement, and one should compare the actual generation quality (In fact, improving perplexity from 27.89 to 26.69 might not necessarily indicate better generation quality). It would be helpful to directly compare generation quality, e.g., comparing rouge-2 of the generations, with/without using the plan.
3. Current table 1 evaluation (edit distance) is done at the sentence/chunk level, which might not be a realistic setup. It would be interesting to compare the quality of generating the complete article.

---

> ### Author Rebuttal · Authors · 2024-05-31
>
> *Methodology*:
> 1. We do not agree that the proposed actions are called **“next sentence embeddings”** as they represent a cluster of sentences at a more abstract level. Consequently, the planning model  does not perform  “next sentence language modeling” for the same reason.
> 2. As discussed in Section 3.1, we posit that it is necessary to compute abstractions that correspond to common writing patterns. By nature, planning in natural language involves a form of abstraction to deal with the vast action space. As Figure 2 demonstrates and we discuss in Section 5.3, there is a **tradeoff between the granularity of actions (and hence informativeness for generation) and the difficulty of predicting the correct action**. In the case that two consecutive sentences are guided by the same action, the language model provides fine-grained control conditioned on previous words and sentences through attention.
> 3. We have tested **your suggested alternative way of using a signal from the language model**, namely in our ablation that encodes the entire context using GPT-2 at once rather than encoding each context sentence with a sentence transformer and then fused into one representation with a trained transformer. We observed that this yielded no improvement (see Table 2 second row and third paragraph of Section 3.2). We will state explicitly that this ablation uses GPT-2 in the final paper.
>
> *Experiments*:
> 1. It is not true that **None** is our only baseline. The main baseline to our proposed method (“Predicted/Predicted” in Table 1) is “Fixed/Fixed”, which finetunes the same architecture on the same data using the regular causal LM objective.
> Our improvement is consistent across 2 models, and more than 1 perplexity point is a moderately sizable difference commonly reported in the LM literature.
> 2. & 3. We now evaluate generation performance via **ROUGE-2** and **edit distance on long articles** up to 2048 tokens. The table below shows the *relative* improvements (in %) of our method over the Fixed baseline, cutting off generation at different points.
>
> | |128|256|512|1024|2048|
> |---|---|---|---|---|---|
> |GPT-2 Edit|6.0|3.1|2.7|1.0|-0.2|
> |GPT-2 R2|12.8|11.0|12.7|12.0|9.8|
> |OLMo Edit|-0.1|1.7|2.6|2.3|2.2|
> |OLMo R2|10.2|15.5|19.2|20.3|15.2|
>
> *Literature*: Thanks for the suggested controlled generation literature. We will include it in our related work section on controlled text generation, where, in contrast to our method, the control codes are provided by the user.

---

> > ### Comment · Reviewer_BMkB · 2024-06-06
> > **Thanks for rebuttal!**
> >
> > Thank you for your responses and clarifications.
> >
> > Overall I like the idea of the framing of this work. Based on the rebuttal and the other reviews, I want to increase my score from 4 to 6. I am not fully convinced with the framing of the “action vector” and some parts of the experiments. I’ll add some quick thoughts below for further discussion for the future, but I do not expect the authors to respond before the rebuttal deadline.
> >
> > > - We do not agree that the proposed actions are called “next sentence embeddings” as they represent a cluster of sentences at a more abstract level. Consequently, the planning model does not perform “next sentence language modeling” for the same reason.
> > > - As discussed in Section 3.1, we posit that it is necessary to compute abstractions that correspond to common writing patterns. By nature, planning in natural language involves a form of abstraction to deal with the vast action space. As Figure 2 demonstrates and we discuss in Section 5.3, there is a tradeoff between the granularity of actions (and hence informativeness for generation) and the difficulty of predicting the correct action. In the case that two consecutive sentences are guided by the same action, the language model provides fine-grained control conditioned on previous words and sentences through attention.
> >
> > - I partly agree with you and I would take back my original statement a bit. But I think the question still exists: starting (even clusters of) sentence representation roots the vector based on the sentence representation space. An alternative way is to purely learn this action as a latent variable model: P(a|x). That might result in a completely different representation space and still might be somewhat helpful. Another alternative is the PEER paper that specifies the editing actions: https://arxiv.org/abs/2208.11663. Regarding the “tradeoff between the granularity of actions (and hence informativeness for generation) and the difficulty of predicting the correct action”, this really boils down to a broader question: if the action is just “vague” and hard to model, then it’s unclear whether it is helpful or in which way it can help the LM.
> >
> > > We have tested your suggested alternative way of using a signal from the language model, namely in our ablation that encodes the entire context using GPT-2 at once rather than encoding each context sentence with a sentence transformer and then fused into one representation with a trained transformer. We observed that this yielded no improvement (see Table 2 second row and third paragraph of Section 3.2). We will state explicitly that this ablation uses GPT-2 in the final paper.
> >
> > - That’s good! Sorry I somehow missed that part when writing the review.

---

> > > ### Author Response · Authors · 2024-06-07
> > > **Thank you for engaging with our rebuttal!**
> > >
> > > We agree that it is preferable to have action representations that can evolve dynamically with the data. For this reason, we let action embeddings adapt when passing them to the LM. Nonetheless, we consider learning better actions from text as one of the top avenues for future work. Thank you for your suggestions in this regard.

---

### Decision · Program_Chairs · 2024-07-10

**Decision:**

Accept

**Comment:**

The paper presents a technique for generating a high level plan of sentence latents, then followed by the text.

The reviewers generally agree on the quality of the modeling and execution, and find the overall idea interesting. The main issues are around the premise that clusters of sentence embeddings represent writing actions, and the evaluation.

I think both these limitations are important.
- There isnt convincing evidence that the clusters represent actions around writing. Rather it is likely content modeling previously explored in generation and summarization literature. In addition, using only data from wikipedia is likely to accentuate this behavior as wikipedia has a very regular topic structure.
- So the evaluation needs to be strengthened with experiments on texts with different types and purpose. Eg. summarization outputs or compare/contrast where writing actions need to be captured in a general way beyond topic. Currently, the perplexity type evaluations only on wikipedia are less convincing. The paper can be strengthened with evaluations on actual tasks/benchmarks with downstream metrics.

[comments from the PCs] Please make sure to follow up on the AC comments and revise as needed.